# Sex Differences among Overweight/Obese Kidney Transplant Recipients Requiring Oxygen Support Amid the COVID-19 Pandemic

**DOI:** 10.3390/medicina59091555

**Published:** 2023-08-27

**Authors:** Alexandre Veronese-Araújo, Débora D. de Lucena, Isabella Aguiar-Brito, Marina P. Cristelli, Hélio Tedesco-Silva, José O. Medina-Pestana, Érika B. Rangel

**Affiliations:** 1Nephrology Division, Department of Medicine, Federal University of São Paulo, São Paulo 04038-031, SP, Brazil; veronese.alexandre@unifesp.br (A.V.-A.); deblucena@yahoo.com.br (D.D.d.L.); isabella.aguiar@unifesp.br (I.A.-B.); heliotedesco@medfarm.hrim.com (H.T.-S.); medina@hrim.com.br (J.O.M.-P.); 2Hospital do Rim, São Paulo 04038-002, SP, Brazil; ninacristelli@yahoo.com.br; 3Hospital Israelita Albert Einstein, São Paulo 05652-900, SP, Brazil

**Keywords:** kidney transplant, COVID-19, outcomes, obesity, overweight, sex differences

## Abstract

*Background and Objectives*: Overweight/obesity puts individuals at greater risk for COVID-19 progression and mortality. We aimed to evaluate the impact of overweight/obesity on oxygen (O_2_) requirement outcomes of male and female kidney transplant recipients (KTRs) during the COVID-19 pandemic. *Materials and Methods*: We conducted a retrospective analysis of a cohort of KTRs diagnosed with COVID-19. Participants were stratified based on BMI categories, and data on the need for O_2_ therapy outcome were collected and analyzed separately for male and female KTRs. *Results*: In total, 284 KTRs (97 males and 187 females) were included in the study. Overweight/obesity was observed in 60.6% of male KTRs and 71% of female KTRs. Strikingly, overweight/obese women had a significantly higher requirement for supplemental O_2_ (63.3% vs. 41.7%, OR = 2.45, *p* = 0.03), particularly among older individuals (OR = 1.05, *p* = 0.04), smokers (OR = 4.55, *p* = 0.03), those with elevated lactate dehydrogenase (LDH) levels (OR = 1.01, *p* = 0.006), and those with lower admission and basal estimated glomerular filtration rate (eGFR) levels. Within this cohort, the necessity for O_2_ supplementation was correlated with more unfavorable outcomes. These included heightened mortality rates, transfers to the intensive care unit, employment of invasive mechanical ventilation, and the emergence of acute kidney injury requiring hemodialysis. On the other hand, although overweight/obese male KTRs had a higher prevalence of hypertension and higher fasting blood glucose levels, no significant association was found with COVID-19-related outcomes when compared to lean male KTRs. *Conclusions*: Overweight/obesity is highly prevalent in KTRs, and overweight/obese women demonstrated a higher need for supplemental O_2_. Therefore, the early identification of factors that predict a worse outcome in overweight/obese female KTRs affected by COVID-19 contributes to risk stratification and guides therapeutic decisions.

## 1. Introduction

Kidney transplant recipients (KTRs) face a significantly higher risk for severe COVID-19 outcomes, leading to increased mortality rates compared to the general population [1]. These concerning findings highlight the importance of comprehending the specific challenges encountered by KTRs during the pandemic. Our group has previously demonstrated that KTRs possess high rates of comorbidities, such as hypertension [2], diabetes mellitus (DM) [3], and overweight/obesity [4]. These comorbidities further compound the susceptibility of KTRs to severe COVID-19 complications, making tailored care strategies essential.

Obesity is an independent risks factor for the severity of COVID-19 [5]. Researchers have highlighted that obesity exacerbates the inflammatory response and hampers lung function, thereby contributing to the manifestation of more severe respiratory symptoms and prolonged viral shedding [5,6].

Obesity is a key component of metabolic syndrome, a cluster of metabolic abnormalities that often occur together and increase the risk of cardiovascular disease and type 2 DM [7]. In the context of metabolic syndrome, obesity contributes to the development of insulin resistance, dyslipidemia, and hypertension. Excess adipose tissue produces hormones and inflammatory molecules that disrupt insulin signaling and promote a chronic low-grade inflammatory state [8]. This combination of factors significantly raises the risk of cardiovascular complications and the development of DM. Therefore, managing obesity is essential in addressing metabolic syndrome and reducing the associated health risks [9].

Additionally, COVID-19 outcomes have shown notable differences between men and women. Several studies have highlighted that men tend to experience more severe illness, including higher rates of hospitalization, intensive care unit admission, and mortality compared to women [10]. Men have been observed to have more severe respiratory symptoms, greater systemic inflammation, and a higher prevalence of comorbidities such as hypertension and cardiovascular disease [10,11]. On the other hand, women appear to mount more robust immune responses, have lower mortality rates, and may exhibit milder manifestations of the disease. Underlying biological, genetic, and hormonal factors likely contribute to these sex-based differences in COVID-19 outcomes [12]. Understanding and addressing these disparities is crucial for developing targeted strategies for prevention, treatment, and care that consider the unique characteristics and susceptibilities of both men and women affected by COVID-19.

In our previous study, we established that body mass index (BMI) was an independent risk factor for oxygen (O_2_) requirement, particularly among overweight/obese KTRs who were older, smokers, and exhibited higher levels of lactate dehydrogenase (LDH), as well as lower levels of estimated glomerular filtration rate (eGFR), lymphocytes, and sodium at admission [4]. Therefore, this follow-up investigation aims to delve deeper into the differences between overweight/obese male and female KTRs, shedding light on crucial risk factors associated with O_2_ requirement. 

## 2. Patients and Methods

### 2.1. Study Design and Setting

We conducted a cohort, cross-sectional, observational, and descriptive study at Hospital do Rim, São Paulo, SP, Brazil. The medical records of patients who were either hospitalized or non-hospitalized with the diagnosis of COVID-19 during the study period of March to August 2020 were evaluated, corresponding to the first wave of COVID-19 in Brazil. We included only patients in whom nasopharyngeal swab RT-PCR (reverse transcriptase-polymerase chain reaction) detected SARS-CoV-2. The population at risk included 11,875 KTRs. Of 590 KTRs who became ill, 284 were included in the study, as previously described [2]. The remaining patients were excluded due to admission to other services. Data were collected regarding patient demographics and laboratory parameters on admission with COVID-19 symptoms. The last patient was included in the study on 30 August 2020. The Ethics and Research Committee of the Federal University of São Paulo (CAEE 35311020.9.0000.8098) approved the study (30 March 2020). Informed consent was obtained from all patients, whereas a waiver was granted for patients who died in other hospitals.

Patient demographics included age, sex, race, body mass index (BMI), time of transplant, type of donor, as well as the presence of comorbidities (hypertension, DM, smoking, chronic obstructive pulmonary disease [COPD], heart disease, liver disease, and autoimmune disease). 

Hypertension was defined whether individuals were on anti-hypertensive drugs, DM was defined according to the use of insulin and/or oral antidiabetics, heart disease whether heart failure and/or coronary artery disease were present, and liver disease whether hepatitis B or C were diagnosed. BMI analysis was performed using the World Health Organization criteria: <25 kg/m^2^ considered normal, ≥25–29.9 kg/m^2^ considered overweight, and ≥30 kg/m^2^ considered obesity. 

All patients were under a steroid regimen, alongside other included medications: tacrolimus (*n* = 230, 81%), mycophenolate (*n* = 171, 60.2%), azathioprine (*n* = 69, 24.3%), cyclosporine (*n* = 34, 12%), mammalian target of rapamycin (mTOR) inhibitors (*n* = 33, 11.6%), and Belatacept (*n* = 1, 0.35%).

### 2.2. Laboratory Testing

On admission, we evaluated in-hospital laboratory data: serum creatinine, glycemia, alanine aminotransferase (ALT), aspartate aminotransferase (AST), lymphocytes, D-dimer, lactate dehydrogenase (LDH), and C-reactive protein (CRP). As for laboratory data before admission, we collected baseline creatinine (mean the last three measurements), fasting blood glucose (FBG, last measurement within 6 months), and glycated hemoglobin (HbA1c, last measurement within 1 year).

The estimated glomerular filtration rate (eGFR) was calculated using the formula defined in the CKD-EPI (Chronic Kidney Disease Epidemiology Collaboration) study.

### 2.3. Statistical Analyses

Two groups of renal recipients affected by COVID-19, e.g., overweight/obesity or BMI ≥ 25 kg/m^2^ and normal weight or BMI < 25 kg/m^2^ were separated. The primary aim was to investigate the differences in demographic and laboratory data, and COVID-19-related outcomes in both males and females who were overweight/obese compared to lean KTRs. The secondary aim was to investigate the risk factors associated with COVID-19-related outcomes, in particular, supplemental oxygen (O_2_) requirement in overweight/obese KTRs. Thus, we performed univariate analyses of demographic and laboratory data, and when the *p*-value was ≤0.1, the variables were entered simultaneously into a binary logistic regression model. The outcomes included death, transfer to intensive care unit (ICU), acute kidney injury (AKI) classified according to KDIGO (Kidney Disease Guidelines), need for hemodialysis (HD), supplemental O_2_, and invasive mechanical ventilation (IMV). Oxygen requirement was defined as any use of oxygen, including nasal prongs, masks, and non-invasive ventilation or high-flow, in particular when patients presented a SatO_2_ < 94% on room air and dyspnea. The results were expressed as odds ratios (ORs) with a confidence interval (CI) of 95%. 

An independent samples *t*-test and chi-square test were used to identify the association between BMI and demographic and laboratory parameters, and the outcomes previously mentioned. Data were described as mean ± standard deviation or median and interquartile range, as indicated. Frequencies and percentages were reported for qualitative data. 

Next, to determine which factors in both male and female KTRs who were overweight/obese increased the risk of the need for supplemental O_2_, demographic, and laboratory data were analyzed using the binary logistic regression model, as previously described. 

Receiver Operating Characteristic (ROC) curves were used to identify the laboratory parameters associated with COVID-19-related outcomes. To calculate the LDH and eGFR cut-off values with better sensitivity and specificity for outcomes, we used the Youden index.

BMI was also evaluated as a continuous variable in a model of linear regression using demographic and laboratory data, as well as those outcomes, for both male and female KTRs.

We analyzed the data using IBM^®^ SPSS (Statistical Product and Services Solutions, version 20.0, SPSS Inc, Chicago, IL, USA). A *p*-value of < 0.05 was considered significant for all data analyses. 

## 3. Results

In our cohort, 60.6% (*n* = 97/160) of male KTRs were classified as overweight/obese, whereas 71% (*n* = 88/124) of female KTRs fell into the overweight/obese category (Appendix A). Among overweight/obese males, a high prevalence of hypertension (86.6% vs. 73%, *p* = 0.03) and elevated FBG before admission (127.9 ± 65.5 vs. 106.4 ± 36.7 mg/dL, *p* = 0.02) were observed (Table 1). Additionally, overweight/obese males were less likely to developed lower grades of AKI less often (stage 1; 6.2% vs. 17.5%, *p* = 0.03). However, no significant differences in outcomes were observed between lean and overweight/obese male KTRs. 

Remarkably, the overweight/obese female population exhibited a higher frequency of requiring supplemental O_2_ compared to their lean counterparts (63.6% vs. 41.7%, odds ratio [OR] = 2.45, *p* = 0.03) (Table 2). 

Within the overweight/obese female group, several factors including age (54.6 ± 11.7 vs. 49.5 ± 9.8 years, *p* = 0.04), smoking (32.1% vs. 9.4%, *p* = 0.03), basal eGFR (44.3 ± 21.4 vs. 56.4 ± 25.9 mL/min/1.73m^2^, *p* = 0.03), on-admission eGFR (32.3 ± 21.3 vs. 45.6 ± 20.1 mL/min/1.73m^2^, *p* = 0.01), and LDH levels (367.3 ± 227.4 vs. 237.5 ± 88.0 U/L, *p* = 0.006) were associated with an increased risk of requiring supplemental O_2_ (Table 3). Multivariate analysis identified age (*p* = 0.02) as an independent risk factor for the need for O_2_ supplementation in this population (Table 3). When we analyzed the outcomes of overweight/obese female KTRs who required O_2_, we observed that these patients exhibited elevated rates of mortality, transfer to ICU, IMV, and AKI necessitating HD (Table 3). 

Furthermore, the ROC curve for LDH revealed an AUC of 0.792 (*p* < 0.0001) with a sensitivity of 70.5% and a specificity of 66.7% for values greater than 237.0 U/L in overweight/obese females requiring O_2_ supplementation (Figure 1A). Additionally, baseline and on-admission eGFR yielded AUCs of 0.354 (*p* = 0.02) and 0.285 (*p* = 0.001) with sensitivities of 46.4% and 34% and specificities of 84.4% and 71.9% for eGFR values lower than 38 mL/min/1.73 m^2^ and 38 mL/min/1.73 m^2^, respectively (Figure 1B). 

To further support our findings, we conducted a linear regression model using BMI as a continuous and dependent variable (Appendix A). Notably, in the analysis of female KTRs, BMI remained a significant risk factor for O_2_ requirement (OR = 2.51, *p* = 0.004; Appendix A). 

## 4. Discussion

In our study, we observed distinct patterns of COVID-19 outcomes between male and female KTRs. Among male KTRs who were overweight/obese, we found a higher prevalence of hypertension and poorer glycemic control. Additionally, they were more likely to develop a more severe stage of AKI. However, we did not observe differences in other COVID-19-related outcomes compared to lean male KTRs. Surprisingly, among overweight/obese female KTRs, we observed a higher frequency of requiring supplemental O_2_. Age, smoking, lower renal function, and higher LDH levels were associated with an increased risk of needing O_2_ in this group. Among overweight/obese women requiring O_2_, elevated rates of mortality, ICU transfers, employment of IMV, and AKI necessitating HD were observed. These findings suggest that overweight/obesity and its associated comorbidities may have a different impact on the kidney transplant setting. 

Obesity, when combined with other underlying cardiometabolic risk factors, such as hypertension, DM, and lipid disorders, can worsen clinical outcomes in the context of COVID-19 [13]. The number of metabolic syndrome components, particularly central obesity, has been associated with increased COVID-19 severity [14]. Obesity itself has been identified as an independent risk factor for SARS-CoV-2 infection (OR = 1.47–2.73), hospitalization (OR = 1.72–2.13), severe disease (OR = 3.81), the need for ICU admission (OR = 1.74–2.25), IMV (OR = 1.66), and mortality (OR = 1.48–1.61) [15,16]. Importantly, a temporal relationship between respiratory failure and kidney injury has been demonstrated in individuals with COVID-19 [17], as exemplified by our cohort of overweight/obese female KTRs requiring O_2_. This correlation is underpinned by analogous pathophysiologic mechanisms within kidney and lung tissues, incorporating direct damage by SARS-CoV-2, localized inflammation, endothelial dysfunction, and the development of microthrombi [18]. As a result, these patients exhibit a more unfavorable outcome [19]. 

However, it should be noted that the association between obesity and worse COVID-19 outcomes is often influenced by the presence of hypertension and DM [20]. 

Obesity is associated with COVID-19 severity through several mechanisms. Firstly, adipose tissue expresses the ACE2, a receptor used by SARS-CoV-2 for cell entry [21]. This suggests that this tissue may serve as a reservoir for viral replication, leading to increased COVID-19 burden. Additionally, ACE2 gene expression is higher in subcutaneous and visceral adipose tissues compared to lung cells in obese patients [22,23], along with elevated levels of soluble ACE2 (sACE2) in patients with metabolic syndrome [24]. Notably, chronic glucocorticoid use in KTRs may also contribute to obesity [25].

Secondly, the inflammatory imbalance in obesity affects the production of adipokines. Proinflammatory adipokines such as leptin are upregulated, while anti-inflammatory adipokines such as adiponectin are downregulated in severe COVID-19 [26]. This abnormality exacerbates cerebrovascular and cardiovascular diseases, insulin resistance, and increases susceptibility to viral lung infections [27]. These factors can ultimately contribute to the need for supplemental O_2_.

SARS-CoV-2 can also dysregulate the immune system, leading to endothelial cell damage associated with thrombosis and inflammation (thromboinflammation) [28] and imbalances in the renin–angiotensin–aldosterone and kallikrein–kinin systems [21]. Ectopic fat accumulation, including epicardial and visceral adipose tissue, contributes to the release of chemokines, cytokines, and adipokines, further driving COVID-19 severity in obese patients [29]. 

Thirdly, obesity-related comorbidities such as heart, kidney, and pancreatic diseases can contribute to dysregulation of the ACE/ACE2 axis in tissue associated with these diseases [21]. Increased ACE2 expression has been observed in overweight patients with COPD [30]. Smoking, which upregulates ACE2 expression in the lungs [31], can also contribute to the need for supplemental O_2_ in overweight/obese KTRs who smoke.

Lastly, higher grades of obesity are associated with lower levels of the PaO_2_/FIO_2_ ratio, indicating a ventilation capacity failure due to mechanical impairment and the burden of COVID-19 [32]. Obese patients experience pulmonary pathophysiological changes, including increased atelectasis formation due to the negative effects of thoracic wall weight and abdominal fat mass [33]. They also exhibit increased work of breathing and abnormalities in blood gas exchange [34]. 

In our study, despite overweight/obese male KTRs exhibiting a higher prevalence of hypertension and elevated glycaemia before admission, they did not experience worse COVID-19-related outcomes compared to lean male KTRs. It is important to highlight that hypertension has been identified as an independent risk factor for COVID-19 progression and mortality [2,35], as well as glycemic levels [3,36] in KTRs and the general population. Nevertheless, the case-fatality rate among KTRs remains higher than that of the general population [37], as we observed in our study (approximately 25%). Additionally, the prevalence of obesity-related comorbidities such as hypertension, DM, and heart disease was similar between overweight/obese male KTRs and their counterparties in our population (86.6% vs. 73%, 40.2% vs. 33.3%, and 13.4% vs. 11.1%, respectively). A similar pattern was observed in overweight/obese female KTRs compared to lean individuals in terms of hypertension (69.3% vs. 63.9%), DM (42% vs. 41.7%), and heart disease (12.5% vs. 2.8%). These findings indicate that obesity and its related comorbidities may have a different impact within the kidney transplant population, which may be explained by the dysregulation of the ACE1/ACE2 axis [21,38]. 

It is worth noting that a majority of our KTR population was overweight/obese, and this finding was consistent among males and females. However, despite this consistency, no discernible differences in mortality rates were observed between lean and overweight/obese male and female KTR individuals. 

Sex differences in non-transplant individuals regarding BMI indicate that females with COVID-19 require hospitalization more frequently, while males have worse outcomes such as ICU transfer, the need for IMV, and higher mortality rates, which can be partly explained by higher rates of hypertension and DM, as mentioned earlier [20]. Moreover, age seems to have a more pronounced effect on COVID-19 outcomes in males, whereas obesity has a stronger bearing on outcomes in females [39,40]. 

Several studies have investigated the impact of sex and age on the levels of ACE2. The higher expression of ACE2 in men and its location on the X chromosome may explain why SARS-CoV-2 binds more frequently to ACE2 in men [38]. In females, ACE2 on either X chromosome can recognize the virus, but the chance of both ACE2 copies binding perfectly to the virus is low, allowing unbound ACE2 to cleave angiotensin II and decrease COVID-19-related complications [41]. Other genes involved in the immune response, located on the X chromosome, may contribute to a more robust immune response in females [42].

Additionally, females mount a stronger adaptive immune response, while males exhibit a more robust innate immune response [43]. In COVID-19 patients, immune profiling has unveiled dynamic alterations in both innate and adaptive immune cells, with specific natural killer-cell receptors and IgM+ B cells linked to significant CD4 and CD8 T-cell exhaustion, thereby elevating the risk of intubation and mortality [44]. These immune dynamics can elucidate the variations in COVID-19 progression between males, with potential implications for clinical outcomes. 

Likewise, estrogens and their receptors play a crucial role in regulating innate immune responses to viral infections by regulating the type-I interferon (IFN) response through several mechanisms, including inhibiting the production of type-I IFNs, and suppressing IFN signaling [12,41]. They can also modulate the production of proinflammatory cytokines by blocking NF-κB signaling [12,41]. In addition to regulating immune responses, estrogens and their receptors also have an impact on the severity of viral infections. These hormones have been shown to mitigate this inflammatory response by reducing the overexpression of proinflammatory cytokines, leading to more balanced immune responses that can effectively clear the virus without causing excessive damage to the host [45]. Estrogen may also regulate the ACE1/ACE2 axis and leads to anti-inflammatory and anti-oxidative effects [12,46]. 

However, the protective effect of the female sex on COVID-19 outcomes, attributed to genetic factors (ACE2 gene location in the X chromosome), immunologic privilege, and hormonal benefits (estrogen) [12], may be compromised by obesity, as we observed in our study. Previous studies have shown higher levels of ACE2 expression in adipose tissue in females compared to males [23], and increased levels of plasma sACE2, released from cytoplasmic membrane to plasma when tissues are damaged [21], were more pronounced in postmenopausal women compared to premenopausal women [24]. Consequently, the beneficial effect of estrogen may be diminished during the postmenopausal period. Although we did not assess the menopausal status in our population, which is likely high given the mean age of 52 years. 

Moreover, in a pre-clinical model of diet-induced obesity in K18-hACE2 transgenic mice, obese female animals demonstrated a shorter time to morbidity, indicating increased susceptibility to SARS-CoV-2, while the obese male mice did not exhibit the same trend. Furthermore, this increased susceptibility in obese female animals was linked to higher viral RNA load and interferon production [47]. Additionally, in non-transplanted individuals, it was discovered that a higher BMI posed a greater risk of COVID-19 mortality in females when compared to males [40]. These observations emphasize the importance of considering gender disparities and the potential impact of obesity on COVID-19 outcomes.

Overall, understanding the influence of sex and age on ACE2 expression and immune responses is crucial for elucidating the differences in COVID-19 outcomes and developing tailored therapies. Moreover, the identification of individuals at high risk of COVID-19-related complications is of paramount importance for effective healthcare management. Emerging data indicate factors such as female sex, pulmonary disease, DM, obesity, and organ transplants as potential risk factor for long COVID-19 [48,49]. It is worth noting that obesity rates have been on the rise in Brazil, particularly among women [50,51]. Therefore, our findings underscore the importance of policymakers, stakeholders, and patients themselves in implementing non-pharmacological, such as lifestyle modifications, and pharmacological approaches to address obesity and related risk factors effectively.

In our study, it is important to highlight that all data were retrospectively collected, and the sample size was limited. Importantly, the attributes of our population, particularly the heightened occurrence of overweight/obesity among KTR women, in conjunction with the limitations imposed by the confined geographical area, could potentially restrict the generalizability of our findings. Additionally, we did not assess the impact of the immunosuppressive regimen on COVID-19 outcomes, a pivotal aspect for consideration. This is particularly relevant as it could be associated with elevated trough levels, possibly stemming from factors such as vomiting, diarrhea, and hepatic injury [52]. Moreover, the impact of longitudinal analyses of laboratory parameters and the vaccination on COVID-19 outcomes in KTRs who are overweight/obese should be further investigated. 

In conclusion, our findings highlight the importance of considering gender disparities and the potential impact of obesity on COVID-19 outcomes in the kidney transplant setting. 

## Figures and Tables

**Figure 1 medicina-59-01555-f001:**
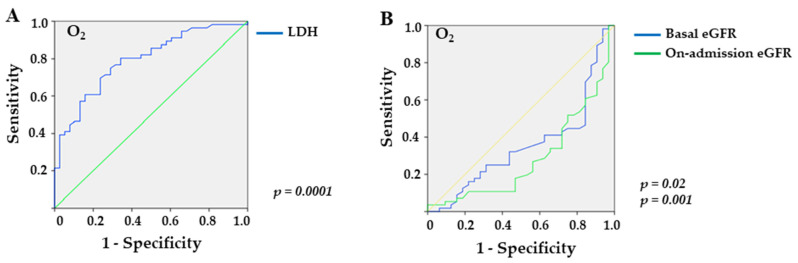
Sensitivity and specificity of laboratory markers in female overweight/obese KTRs with COVID-19 who required supplemental O_2_: (**A**) LDH (lactate dehydrogenase; *p* = 0.0001), and (**B**) basal eGFR (estimated glomerular filtration rate; *p* = 0.02) and on-admission eGFR (*p* = 0.001).

**Table 1 medicina-59-01555-t001:** Clinical and epidemiological characteristics, laboratory data, and outcomes of male kidney transplant recipients who were overweight/obese and lean.

Variables	BMI ≥ 25, Male (*n* = 97, 60.6%)	BMI < 25, Male (*n* = 63, 39.4%)	TOTAL, Male (*n* = 160, 100%)	Univariate Analysis OR (95% CI)
Age (years)	53.8 ± 10.9	52.2 ± 14.5	53.2 ± 12.4	1.01 (0.98–1.04, *p* = 0.43)
Race (*n*, %) White Black/brown	62 (63.9) 35 (36.1)	34 (54.0) 29 (46.0)	96 (60.0) 64 (40.0)	1.51 (0.79–2.88, *p* = 0.21)
Transplant time (months)	94.1 [31.0;145.0]	95.6 [39.0;129.0]	94.7 [33.8;142.3]	1.00 (0.99–1.00, *p* = 0.90)
Donor type (*n*, %) Live Deceased	35 (36.1) 62 (63.9)	15 (23.8) 48 (76.2)	50 (31.3) 110 (68.7)	1.81 (0.89–3.68, *p* = 0.10)
Hypertension (*n*, %)	84 (86.6)	46 (73.0)	130 (81.3)	2.39 (1.07–5.35, ***p* = 0.03**)
Diabetes mellitus (*n*, %)	39 (40.2)	21 (33.3)	60 (37.5)	1.35 (0.69–2.61, *p* = 0.38)
COPD (*n*, %)	3 (3.1)	2 (3.2)	5 (3.1)	0.97 (0.16–6.00, *p* = 0.10)
Heart disease (*n*, %)	13 (13.4)	7 (11.1)	20 (12.5)	1.24 (0.46–3.30, *p* = 0.67)
Neoplasia (*n*, %)	8 (8.2)	6 (9.5)	14 (8.8)	0.85 (0.28–2.59, *p* = 0.78)
Liver disease (*n*, %)	4 (4.1)	3 (4.8)	7 (4.4)	0.86 (0.190–3.98, *p* = 0.85)
Autoimmune disease (*n*, %)	0 (0)	1 (1.6)	1 (0.1)	0.00 (0.0001, *p* = 1.00)
Smoking (*n*, %)	22 (22.7)	13 (20.6)	35 (21.9)	1.29 (0.58–2.85, *p* = 0.54)
**Laboratory data**				
Basal eGFR	49.7 ± 22.9	50.4 ± 17.3	50.0 ± 24.6	0.99 (1.00–1.01, *p* = 0.87)
Admission eGFR	38.2 ± 19.8	37.3 ± 23.2	37.8 ± 21.1	1.00 (1.00–1.02, *p* = 0.78)
Previous glucose (mg/dL)	127.9 ± 65.5	106.4 ± 36.7	119.4 ± 56.7	1.01 (1.00–1.01, ***p* = 0.02**)
Admission glucose (mg/dL)	199.7 ± 112.9	147.2 ± 88.7	180.9 ± 107.3	1.01 (1.00–1.01, *p* = 0.06)
Previous Hb1Ac (%)	7.0 ± 2.0	6.6 ± 2.0	6.8 ± 2.0	1.11 (0.92–1.34, *p* = 0.22)
CRP (mg/dL)	9.0 [1.9; 13.3]	9.0 [2.4; 14.6]	9.0 [1.9; 13.6]	1.00 (0.96–1.04, *p* = 0.96)
LDH (U/L)	359.0 [244.0; 441.0]	313.0 [228.0; 361.5]	339.7 [236.5; 402.5]	1.00 (1.00–1.00, *p* = 0.13)
Lymphocytes (mm^3^)	903.5 [458.0; 1067.0]	896.7 [474.3; 1103.0]	901.0 [460.0; 1094.0]	1.00 (1.00–1.00, *p* = 0.96)
D-dimer (µg/L)	2.1 [0.6; 1.8]	2.5 [0.5; 2.2]	2.3 [0.6; 1.9]	0.98 (1.00–1.07, *p* = 0.61)
AST (U/L)	37.9 [20.0; 41.0]	36.2 [23.0; 43.0]	37.3 [20.0; 42.3]	1.00 (1.00–1.02, *p* = 0.73)
ALT (U/L)	31.9 [15.0; 36.0]	27.6 [17.8; 33.8]	30.3 [15.0; 36.0]	1.01 (1.00–1.02, *p* = 0.38)
Sodium (mEq/L)	135.6 ± 3.8	135.0 ± 5.2	135.4 ± 4.4	1.03 (0.95–1.12, *p* = 0.42)
**Outcomes**				
Death (*n*, %)	36 (37.1)	16 (25.4)	52 (32.5)	1.73 (0.86–3.50, *p* = 0.12)
ICU (*n*, %)	47 (48.5)	29 (46.0)	76 (47.5)	1.10 (0.58–2.08, *p* = 0.76)
O_2_ (*n*, %)	52 (53.6)	30 (47.6)	82 (51.3)	1.27 (0.67–2.40, *p* = 0.46)
IMV (*n*, %)	42 (43.3)	19 (30.2)	61 (38.1)	1.77 (0.90–3.46, *p* = 0.10)
AKI (*n*, %)	51 (52.6)	40 (63.5)	91 (56.9)	0.64 (0.33–1.22, *p* = 0.17)
Stage 1	6 (6.2)	11 (17.5)	17 (10.6)	0.31 (0.11–0.89, ***p* = 0.03**)
Stage 2	3 (3.1)	5 (7.9)	8 (5.0)	0.37 (0.08–1.61, *p* = 0.18)
Stage 3	42 (43.3)	24 (38.1)	66 (41.3)	1.24 (0.65–2.37, *p* = 0.51)
HD (*n*, %)	41 (42.3)	23 (36.5)	64 (40.0)	1.27 (0.66–2.44, *p* = 0.47)

BMI: body mass index in kg/m^2^; COPD: chronic obstructive pulmonary disease; eGFR: estimated glomerular rate, in mL/min/1.73 m^2^; Hb1Ac: glycated hemoglobin; CRP: C-reactive protein; LDH: lactate dehydrogenase; AST: aspartate aminotransferase; ALT: alanine aminotransferase; ICU: intensive care unit; O_2_: use of supplemental oxygen; IMV: invasive mechanical ventilation; AKI: acute kidney injury; HD: hemodialysis. All variables are means ± SD, except the variables transplant time, CRP, LDH, lymphocytes, D-dimer, AST, and ALT, which are medians and IQR. OR: odds ratio; 95% CI: 95% confidence interval.

**Table 2 medicina-59-01555-t002:** Clinical and epidemiological characteristics, laboratory data, and outcomes of female kidney transplant recipients who were overweight and lean.

Variables	BMI ≥ 25, Female (*n* = 88, 71.0%)	BMI < 25, Female (*n* = 36, 29.0%)	TOTAL, Female (*n* = 124, 100%)	Univariate Analysis OR (95% CI)
**Age (years)**	**52.8 ± 11.2**	**48.7 ± 13.1**	**51.6 ± 11.9**	1.03 (1.00–1.07, *p* = 0.09)
Race (*n*, %) White Black/brown	59 (67.0) 29 (33.0)	18 (50.0) 18 (50.0)	77 (62.1) 47 (37.9)	2.03 (0.93–4.48, *p* = 0.08)
Transplant time (months)	90.0 [37.8;145.0]	94.7 [26.0;127.3]	91.4 [32.3;132.0]	0.99 (0.99–1.00, *p* = 0.74)
Donor type (*n*, %) Live Deceased	20 (22.7) 68 (77.3)	10 (27.8) 26 (72.2)	30 (24.2) 94 (75.8)	0.77 (0.32–1.85, *p* = 0.55)
Hypertension (*n*, %)	61 (69.3)	23 (63.9)	84 (67.7)	1.28 (0.56–2.89, *p* = 0.56)
Diabetes mellitus (*n*, %)	37 (42.0)	15 (41.7)	52 (41.9)	1.02 (0.46–2.23, *p* = 0.97)
COPD (*n*, %)	3 (3.4)	1 (2.8)	4 (3.2)	1.24 (0.12–12.29, *p* = 0.86)
Heart disease (*n*, %)	11 (12.5)	1 (2.8)	12 (9.7)	5.00 (0.62–40.25, *p* = 0.13)
Neoplasia (*n*, %)	5 (5.7)	2 (5.6)	7 (5.6)	1.02 (0.19–5.54, *p* = 0.98)
Liver disease (*n*, %)	2 (2.3)	0 (0)	2 (1.6)	(0.0001, *p* = 0.99)
Autoimmune disease (*n*, %)	7 (8.0)	0 (0)	7 (5.6)	(0.0001, *p* = 0.99)
Smoking (*n*, %)	21 (29.9)	3 (8.3)	24 (19.4)	3.57 (0.97–13.08, *p* = 0.05)
**Laboratory data**				
Basal eGFR	48.7 ± 23.7	43.3 ± 21.9	47.2 ± 23.3	1.01 (0.99–1.03, *p* = 0.24)
Admission eGFR	37.1 ± 21.7	33.7 ± 25.6	36.1 ± 22.9	1.01 (1.00–1.02, *p* = 0.45)
Previous glucose (mg/dL)	123.0 ± 65.8	117.7 ± 101.5	121.4 ± 77.6	1.00 (1.00–1.01, *p* = 0.73)
Admission glucose (mg/dL)	166.0 ± 99.0	166.9 ± 109.0	166.3 ± 101.0	1.00 (0.99–1.01, *p* = 0.97)
Previous Hb1Ac (%)	7.1 ± 2.0	6.5 ± 1.1	7.0 ± 2.0	1.21 (0.90–1.60, *p* = 0.20)
CRP (mg/dL)	8.6 [2.3; 12.5]	6.6 [1.7; 8.0]	8.0 [2.2; 11.0]	1.03 (0.97–1.09, *p* = 0.30)
LDH (U/L)	321.5 [213.0; 355.0]	317.0 [182.8; 386.0]	320.2 [209.3; 377.5]	1.00 (1.00–1.00, *p* = 0.92)
Lymphocytes (mm^3^)	1016.7 [551.0; 1508.5]	1015.3 [494.8; 1153.8]	1016.3 [523.5; 1298.5]	1.00 (1.00–1.00, *p* = 0.99)
D-dimer (µg/L)	2.2 [0.6; 2.4]	3.1 [0.8; 3.1]	2.4 [0.6; 2.5]	0.91 (0.81–1.03, *p* = 0.15)
AST (U/L)	40.9 [21.0; 38.0]	40.4 [21.5; 40.0]	40.7 [21.0; 40.0]	1.00 (0.99–1.01, *p* = 0.97)
ALT (U/L)	34.3 [14.8; 28.0]	23.7 [14.0; 26.8]	31.1 [14.0; 27.0]	1.01 (1.00–1.02, *p* = 0.48)
Sodium (mEq/L)	134. 6 ± 5.8	134.1 ± 5.5	134.5 ± 5.7	1.02 (0.95–1.09, *p* = 0.66)
**Outcomes**				
Death (*n*, %)	23 (26.1)	9 (25.0)	32 (25.8)	1.06 (0.43–2.59, *p* = 0.90)
ICU (*n*, %)	45 (51.1)	13 (36.1)	58 (46.8)	1.85 (0.83–4.11, *p* = 0.13)
O_2_ (*n*, %)	56 (63.6)	15 (41.7)	71 (57.3)	2.45 (1.11–5.41, ***p* = 0.03**)
IMV (*n*, %)	28 (31.8)	8 (22.2)	36 (29.0)	1.63 (0.66–4.04, *p* = 0.29)
AKI (*n*, %)	52 (59.1)	22 (61.1)	74 (59.7)	0.92 (0.42–2.03, *p* = 0.83)
Stage 1	15 (17.0)	5 (13.9)	20 (16.1)	1.27 (0.43–3.81, *p* = 0.66)
Stage 2	6 (6.8)	0 (0)	6 (4.8)	(0.0001, *p* = 0.99)
Stage 3	31 (35.2)	17 (47.2)	48 (38.7)	0.61 (0.28–1.33, *p* = 0.21)
HD (*n*, %)	29 (33.0)	12 (33.3)	41 (33.1)	0.98 (0.43–2.24, *p* = 0.97)

BMI: body mass index in kg/m^2^; COPD: chronic obstructive pulmonary disease; eGFR: estimated glomerular rate, in mL/min/1.73 m^2^; Hb1Ac: glycated hemoglobin; CRP: C-reactive protein; LDH: lactate dehydrogenase; AST: aspartate aminotransferase; ALT: alanine aminotransferase; ICU: intensive care unit; O_2_: use of supplemental oxygen; IMV: invasive mechanical ventilation; AKI: acute kidney injury; HD: hemodialysis. All variables are means ± SD, except the variables transplant time, CRP, LDH, lymphocytes, D-dimer, AST, and ALT, which are medians and IQR. OR: odds ratio; 95% CI: 95% confidence interval.

**Table 3 medicina-59-01555-t003:** Risk factors for the use of supplemental oxygen (O_2_) and outcomes in female kidney transplant recipients who were overweight/obese.

Variables	O_2_, Female (*n* = 56, 63.6%)	No O_2_, Female (*n* = 32, 36.4%)	Univariate Analysis OR (95% CI)	Multivariate Analysis OR (95% CI)
Age (years)	54.6 ± 11.7	49.5 ± 9.8	1.05 (1.00–1.09, ***p* = 0.04**)	1.07 (1.01–1.13, ***p* = 0.02**)
Race (*n*, %) White Black/brown	39 (69.6) 17 (30.4)	20 (62.5) 12 (37.5)	1.38 (0.55–3.44, *p* = 0.49)	
Transplant time (months)	82.0 ± 68.0	103.9 ± 69.5	0.99 (1.00–1.00, *p* = 0.15)	
Donor type (*n*, %) Live Deceased	9 (16.1) 47 (83.9)	11 (34.4) 21 (65.6)	2.74 (1.00–7.60, *p* = 0.05)	0.47 (0.12–1.91, *p* = 0.29)
Hypertension (*n*, %)	39 (69.6)	22 (68.8)	1.04 (0.41–2.67, *p* = 0.93)	
Diabetes mellitus (*n*, %)	26 (46.4)	11 (34.4)	1.66 (0.67–4.06, *p* = 0.27)	
COPD (*n*, %)	2 (3.6)	1 (3.1)	1.15 (0.10–13.18, *p* = 0.91)	
Heart disease (*n*, %)	7 (12.5)	4 (12.5)	1.00 (0.27–3.72, *p* = 1.000)	
Neoplasia (*n*, %)	3 (5.6)	2 (6.3)	0.85 (0.13–5.37, *p* = 0.86)	
Liver disease (*n*, %)	2 (3.6)	0 (0)	(0.0001, *p* = 0.99)	
Autoimmune disease (*n*, %)	5 (8.9)	2 (6.3)	(0.27–8.06, *p* = 0.66)	
Smoking (*n*, %)	18 (32.1)	3 (9.4)	4.55 (1.19–17.42, ***p* = 0.03**)	3.82 (0.93–15.64, *p* = 0.06)
**Laboratory data**				
Basal eGFR	44.3 ± 21.4	56.4 ± 25.9	0.98 (0.96–1.00, ***p* = 0.03**)	0.99 (0.95–1.03, *p* = 0.64)
Admission eGFR	32.3 ± 21.3	45.6 ± 20.1	0.97 (0.95–0.99, ***p* = 0.01**)	0.97 (0.93–1.01, *p* = 0.15)
Previous glucose (mg/dL)	105.0 [85.5; 152.5]	89.0 [32.0; 174.5]	1.01 (1.00–1.02, *p* = 0.06)	1.01 (1.00–1.03, *p* = 0.08)
Admission glucose (mg/dL)	140.0 [95.0; 174.5]	143.5 [106.8; 250.0]	1.00 (0.99–1.01, *p* = 0.98)	
Previous Hb1Ac (%)	7.3 ± 2.0	6.7 ± 2.0	1.10 (0.89–1.61, *p* = 0.24)	
CRP (mg/dL)	9.5 [3.6; 13.0]	7.3 [1.4; 9.2]	1.03 (0.97–1.10, *p* = 0.29)	
LDH (U/L)	367.3 [231.5; 426.0]	237.5 [165.3; 295.0]	1.01 (1.00–1.01, ***p* = 0.006**)	1.01 (1.002–1016, *p* = 0.17)
Lymphocytes (mm^3^)	941.9 [512.8; 1242.8]	1156.0 [632.0; 1569.0]	0.99 (1.00–1.00, *p* = 0.14)	
D-dimer (µg/L)	2.2 [0.7; 2.4]	2.1 [0.5; 2.4]	1.00 (0.83–1.22, *p* = 0.97)	
AST (U/L)	45.3 [20.5; 39.5]	32.3 [21.5; 35.0]	1.01 (1.00–1.03, *p* = 0.37)	
ALT (U/L)	40.0 [14.0; 28.0]	23.2 [15.0; 27.0]	1.01 (1.00–1.03, *p* = 0.42)	
Sodium (mEq/L)	134.0±6.7	135.7±3.8	0.94 (0.86–1.04, *p* = 0.24)	
**Outcomes**				
Death (*n*, %)	23 (41.1)	0 (0.0)	21. 61 (2.75–169.74, ***p* = 0.003**)	0.45 (0.01–23.78, *p* = 0.69)
ICU (*n*, %)	41 (73.2)	4 (12.5)	19.13 (5.75–63.72, ***p* = 0.0001**)	6.13 (1.38–27.14, ***p* = 0.02**)
IMV (*n*, %)	28 (50.0)	0 (0.0)	31.00 (3.96–242.99, ***p* = 0.001**)	3.68 (0.05–257.65, *p* = 0.55)
AKI (*n*, %)	44 (78.6)	8 (25.0)	11.00 (3.95–30.61, ***p* = 0.0001**)	2.90 (0.81–10.34, *p* = 0.10)
Stage 1	10 (17.9)	5 (15.6)	1.17 (0.36–3.80, *p* = 0.80)	
Stage 2	4 (7.1)	2 (6.3)	1.15 (0.20–6.68, *p* = 0.87)	
Stage 3	30 (53.6)	1 (3.1)	35.77 (4.56–280.48, ***p* = 0.001**)	9.54 (0.39–231.49, *p* = 0.17)
HD (*n*, %)	29 (51.8)	0 (0.0)	33.30 (4.25–261.02, ***p* = 0.001**)	0.56 (0.11–29.06, *p* = 0.77)

BMI: body mass index in kg/m^2^; COPD: chronic obstructive pulmonary disease; eGFR in mL/min/1.73 m^2^; Hb1Ac: glycated hemoglobin; CRP: C-reactive protein; LDH: lactate dehydrogenase; AST: aspartate aminotransferase; ALT: alanine aminotransferase; ICU: intensive care unit; O_2_: use of supplemental oxygen; IMV: invasive mechanical ventilation; AKI: acute kidney injury; HD: hemodialysis. All variables are means ± SD, except the variables transplant time, previous and admission glucose, CRP, LDH, lymphocytes, D-dimer, AST, and ALT, which are medians and IQR. OR: odds ratio; 95% CI: 95% confidence interval.

## Data Availability

The data presented in this study are available on request from the corresponding authors. The data are not publicly available due to clinical patient information.

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
