# Peer review of "Sex Differences among Overweight/Obese Kidney Transplant Recipients Requiring Oxygen Support Amid the COVID-19 Pandemic"

_medicina, 2023, doi:10.3390/medicina59091555_

Round 1

Reviewer 1 Report

dear authors

congratulations for your study

The design is not completely fine and I miss some points:

It is a pity that the number of your cases is short because half of your cases have not been recorded.

In my opinion the division of the study groups, according to BDI, is not adequate because  overweighted  and obesity patients are included in the same group,being very different patients,which could modificate the results

I think that you should point out in the conclusions the fact that the group of overweigthed patients have the same survival that lean patients , being male or female..

besides , the resuls of male and female patients should be compared.

Author Response

We extend our gratitude to the reviewers for their valuable comments, which have undoubtedly played a crucial role in enhancing the manuscript. We have taken all the provided feedback into careful consideration and have made necessary revisions accordingly.

Reviewer 1

dear authors

congratulations for your study

The design is not completely fine and I miss some points:

  • It is a pity that the number of your cases is short because half of your cases have not been recorded.

Response: We appreciate the reviewer's perspective. Regrettably, during the initial wave of the COVID-19 pandemic, our hospital faced significant overcrowding, leading patients to seek medical assistance closer to their residences. Given the vast expanse of the São Paulo city, Brazil, mobility posed challenges during that period.

  • In my opinion the division of the study groups, according to BDI, is not adequate because  overweighted  and obesity patients are included in the same group,being very different patients,which could modificate the results

Response: We appreciate your raising this topic for discussion. We concur with the reviewer's observation that overweight individuals exhibit distinct risks for COVID-19 severity in comparison to obese individuals. Furthermore, among the obese population, substantial disparities exist based on the degree of obesity (Wu J et al, J Intern Med, 2020; Lighter J et al, Clin Infect Dis, 2020; Simonnet A et al, Obesity, 2020). In light of these variations, we have chosen to combine overweight and obese individuals into a single group for analysis. This decision stems from the lower count of obese individuals compared to their overweight counterparts. We have included this information in a new table, which can be found in the supplemental file (Table S1).

                                 Male                       Female             Total   Univariate analysis   

Overweight (n, %)

64 (40.0)

56 (45.2)

120 (42.3)

1.24 (0.77-1.99, p=0.38)

Obesity I (n, %)

27 (16.9)

23 (18.5)

50 (17.6)

1.12 (0.61-2.07, p=0.71)

Obesity II (n, %)

2 (1.3)

7 (5.6)

9 (3.2)

4.73 (0.96-23.17, p=0.06)

Obesity III (n, %)

4 (2.5)

2 (1.6)

6 (2.1)

0.64 (0.12-3.55, p=0.61)

  • I think that you should point out in the conclusions the fact that the group of overweigthed patients have the same survival that lean patients , being male or female.

Response: Thank you for your valuable insight. We absolutely agree that it is essential to highlight in the conclusions that the survival rate of overweight patients is comparable to that of lean patients, regardless of gender. We will ensure to include this crucial point in our conclusions to provide a comprehensive understanding of the study's findings (please see page 9). Your suggestion is greatly appreciated and will contribute to the clarity and significance of our conclusions.

  • besides , the resuls of male and female patients should be compared.

Response: We appreciate your input. To address your recommendation, we have incorporated a comparative analysis of the results for male and female patients. This comparison has been documented in a newly created table, accessible in the supplemental file (Table S1). Notably, the primary distinction observed was a greater prevalence of hypertension among males (81.3% vs. 67.7%, p=0.01), while females exhibited a higher frequency of autoimmune diseases (5.6% vs. 0.1%, p=0.04). We are grateful for your suggestion, as it has enriched the comprehensive understanding of our study's outcomes.

Sincerely, 

Érika B Rangel, MD, PhD

Reviewer 2 Report

The manuscript Sex differences among overweight/obese kidney transplant recipients requiring oxygen support amid the COVID-19 pandemic by Alexandre Veronese-Araújo et al. is a single-center, cross-sectional cohort study aimed at identifying predictive factors in overweight and obese transplant recipients with COVID-19. Their study shows a high prevalence of overweight/obesity among transplant recipients with COVID-19 in their cohort, with overweight/obese women having a more increased need for supplemental oxygen. Understanding and acknowledging sex and gender disparities in transplant recipients affected by COVID-19 is critical for improving patient risk stratification, management, and outcomes. Hence, the relevance of this study. The manuscript is well-written and methodologically sound. Here are some minor comments:

In their study cohort, the authors describe characteristics associated with the kidney transplant (transplant time and donor type) and consider them when performing univariate and multivariate analysis. However, their analysis did not describe or consider the immunosuppressive regime of the patients included in the study. Patients affected by COVID-19 and on immunosuppression have higher blood concentrations of tacrolimus and sirolimus (10.1016/j.ekir.2021.07.012)(10.1016/j.transproceed.2021.01.002). Moreover, a study by Krista Mecadon et al. showed that organ transplant recipients with higher tacrolimus troughs on admission for COVID-19 (> 10 ng/mL) were more likely to require supplemental oxygen (10.1177/10600280221078983). Thus, the manuscript would benefit from adding the immunosuppressant regime of their cohort, at least to patient demographics, and from a more in-depth discussion on the potential role of immunosuppressants in the context of their findings. 

Although in the methods section, the authors mention that the results were expressed as odds ratios (ORs) with a confidence interval (CI) of 95%. To improve the readability of the tables, I suggest adding OR (95% CI) to the headings of each table where appropriate. 

Please define IMV when it first appears in the text. 

References:

McMinn J, Black H, Harrison LL, Geddes C. SARS-CoV-2 and Tacrolimus Blood Concentration in Kidney Transplant Recipients. Kidney Int Rep. 2021 Oct;6(10):2694-2697. doi: 10.1016/j.ekir.2021.07.012.

Hardesty A, Pandita A, Vieira K, Rogers R, Merhi B, Osband AJ, Aridi J, Shi Y, Bayliss G, Cosgrove C, Gohh R, Morrissey P, Beckwith CG, Farmakiotis D. Coronavirus Disease 2019 in Kidney Transplant Recipients: Single-Center Experience and Case-Control Study. Transplant Proc. 2021 May;53(4):1187-1193. doi: 10.1016/j.transproceed.2021.01.002. 

Mecadon K, Hardesty A, Vieira K, Rogers R, Merhi B, Osband AJ, Bayliss Md G, Gohh R, Morrissey P, Farmakiotis D. Elevated Tacrolimus Levels at Time of Diagnosis of COVID-19 Compared to Baseline Among Hospitalized Organ Transplant Recipients. Ann Pharmacother. 2022 Feb 18:10600280221078983. doi: 10.1177/10600280221078983.

Author Response

We extend our gratitude to the reviewers for their valuable comments, which have undoubtedly played a crucial role in enhancing the manuscript. We have taken all the provided feedback into careful consideration and have made necessary revisions accordingly.

Reviewer 2

The manuscript Sex differences among overweight/obese kidney transplant recipients requiring oxygen support amid the COVID-19 pandemic by Alexandre Veronese-Araújo et al. is a single-center, cross-sectional cohort study aimed at identifying predictive factors in overweight and obese transplant recipients with COVID-19. Their study shows a high prevalence of overweight/obesity among transplant recipients with COVID-19 in their cohort, with overweight/obese women having a more increased need for supplemental oxygen. Understanding and acknowledging sex and gender disparities in transplant recipients affected by COVID-19 is critical for improving patient risk stratification, management, and outcomes. Hence, the relevance of this study. The manuscript is well-written and methodologically sound. Here are some minor comments:

  • In their study cohort, the authors describe characteristics associated with the kidney transplant (transplant time and donor type) and consider them when performing univariate and multivariate analysis. However, their analysis did not describe or consider the immunosuppressive regime of the patients included in the study. Patients affected by COVID-19 and on immunosuppression have higher blood concentrations of tacrolimus and sirolimus (10.1016/j.ekir.2021.07.012)(10.1016/j.transproceed.2021.01.002). Moreover, a study by Krista Mecadon et al. showed that organ transplant recipients with higher tacrolimus troughs on admission for COVID-19 (> 10 ng/mL) were more likely to require supplemental oxygen (10.1177/10600280221078983). Thus, the manuscript would benefit from adding the immunosuppressant regime of their cohort, at least to patient demographics, and from a more in-depth discussion on the potential role of immunosuppressants in the context of their findings. 

Response: Thank you for bringing attention to this aspect. All patients were under a steroid regimen, alongside other included medications: Tacrolimus (n=230, 81%), Mycophenolate (n=171, 60.2%), Azathioprine (n=69, 24.3%), Cyclosporine (n=34, 12%), Mammalian Target of Rapamycin (mTOR) inhibitors (n=33, 11.6%), and Belatacept (n=1, 0.35%). Regrettably, detailed levels are unavailable. We have addressed this particular point in the "Limitation paragraph" (page 10) and in the “Methodology” section (page 3) of the study.

  • Although in the methods section, the authors mention that the results were expressed as odds ratios (ORs) with a confidence interval (CI) of 95%. To improve the readability of the tables, I suggest adding OR (95% CI) to the headings of each table where appropriate. 

Response: Thank you for your suggestion. We modified the tables accordingly.  

  • Please define IMV when it first appears in the text. 

Response:  We defined IMV on page 3 when it first appeared.

References:

McMinn J, Black H, Harrison LL, Geddes C. SARS-CoV-2 and Tacrolimus Blood Concentration in Kidney Transplant Recipients. Kidney Int Rep. 2021 Oct;6(10):2694-2697. doi: 10.1016/j.ekir.2021.07.012.

Hardesty A, Pandita A, Vieira K, Rogers R, Merhi B, Osband AJ, Aridi J, Shi Y, Bayliss G, Cosgrove C, Gohh R, Morrissey P, Beckwith CG, Farmakiotis D. Coronavirus Disease 2019 in Kidney Transplant Recipients: Single-Center Experience and Case-Control Study. Transplant Proc. 2021 May;53(4):1187-1193. doi: 10.1016/j.transproceed.2021.01.002. 

Mecadon K, Hardesty A, Vieira K, Rogers R, Merhi B, Osband AJ, Bayliss Md G, Gohh R, Morrissey P, Farmakiotis D. Elevated Tacrolimus Levels at Time of Diagnosis of COVID-19 Compared to Baseline Among Hospitalized Organ Transplant Recipients. Ann Pharmacother. 2022 Feb 18:10600280221078983. doi: 10.1177/10600280221078983.

Sincerely, 

Érika B Rangel, MD, PhD

Reviewer 3 Report

Veronese Araujo A et al evaluated the impact of overweight/obesity on oxygen (O2) requirement outcomes of male and female kidney transplant recipients (KTRs) during the COVID-19 pandemic.

284 KTRs (97 males and 187 females) were included in the study.

The Authors concluded that overweight/obesity, highly prevalent in KTRs women, required a higher need for supplemental O2 compared with the normalweight female KTR subjects.

The work is original and interesting in its conclusions regarding the need for supplemental O2 therapy in obese women.

I have a few comments to make regarding the Results and the Discussion sections:

- Results: Tab 3 is not complete, it would be very interesting to also have the outcome data (as in Tab 1 and 2), separated by oxygen supplementation, of the female population with BMI>25. Any difference in mortality, incidence of AKI, need of HD, and so on, between the 2 groups?

- The discussion is too long and wordy, including observations that are not pertinent to the presented data.

- the characteristics of the population (high incidence of obesity among women with KTR), and the limited geographical studied area may limit the generalization of the message coming out from the presented data.

- as a final comment, it s important to underline that the obese/older female, in comparison with younger/normal weightted had not a higher mortality rate, even if they required a significant high rate of O2 supplementation

good quality

Author Response

We extend our gratitude to the reviewers for their valuable comments, which have undoubtedly played a crucial role in enhancing the manuscript. We have taken all the provided feedback into careful consideration and have made necessary revisions accordingly.

Veronese Araujo A et al evaluated the impact of overweight/obesity on oxygen (O2) requirement outcomes of male and female kidney transplant recipients (KTRs) during the COVID-19 pandemic.

284 KTRs (97 males and 187 females) were included in the study.

The Authors concluded that overweight/obesity, highly prevalent in KTRs women, required a higher need for supplemental O2 compared with the normalweight female KTR subjects.

The work is original and interesting in its conclusions regarding the need for supplemental O2 therapy in obese women.

I have a few comments to make regarding the Results and the Discussion sections:

  • Results: Tab 3 is not complete, it would be very interesting to also have the outcome data (as in Tab 1 and 2), separated by oxygen supplementation, of the female population with BMI>25. Any difference in mortality, incidence of AKI, need of HD, and so on, between the 2 groups?

Response: We appreciate your valuable suggestion. Subsequent to your input, we conducted the requisite statistical analyses. Our findings highlighted that overweight/obese female kidney transplant recipients (KTRs) exhibited more unfavorable outcomes. We have now incorporated this pertinent information into "Abstract" section and Table 3 on Page 6. Furthermore, we have delved into a comprehensive discussion of these findings on Page 9.

Outcomes

O2, female

(n=56, 63.6%)

No O2, female

(n=32, 36.4%)

Univariate analysis

Multivariate analysis

Death (n, %)

23 (41.1)

0 (0.0)

21. 61 (2.75-169.74, p=0.003)

0.45 (0.01-23.78, p=0.69)

ICU (n, %)

41 (73.2)

4 (12.5)

19.13 (5.75-63.72, p=0.0001)

6.13 (1.38-27.14, p=0.02)

IMV (n, %)

28 (50.0)

0 (0.0)

31.00 (3.96-242.99, p=0.001)

3.68 (0.05-257.65, p=0.55)

AKI (n, %)

44 (78.6)

8 (25.0)

11.00 (3.95-30.61, p=0.0001)

2.90 (0.81-10.34, p=0.10)

Stage 1

10 (17.9)

5 (15.6)

1.17 (0.36-3.80, p=0.80)

Stage 2

4 (7.1)

2 (6.3)

1.15 (0.20-6.68, p=0.87)

Stage 3

30 (53.6)

1 (3.1)

35.77 (4.56-280.48, p=0.001)

9.54 (0.39-231.49, p=0.17)

HD (n, %)

29 (51.8)

0 (0.0)

33.30 (4.25-261.02, p=0.001)

0.56 (0.11-29.06, p=0.77)

  • The discussion is too long and wordy, including observations that are not pertinent to the presented data.

Response: We reviewed the “Discussion” section accordingly.   

  • the characteristics of the population (high incidence of obesity among women with KTR), and the limited geographical studied area may limit the generalization of the message coming out from the presented data.

Response: We appreciate your valuable observation. To address this concern, we have integrated this limitation into the manuscript on page 10.

4) as a final comment, it s important to underline that the obese/older female, in comparison with younger/normal weightted had not a higher mortality rate, even if they required a significant high rate of O2 supplementation

Response: Your input is appreciated. We have taken action to address this aspect on page 9 of the manuscript.

Round 2

Reviewer 1 Report

dear authors

thank you for your interest in improving your manuscript.I think that it has improved enough to be published

none

Reviewer 3 Report

I have no further comments

good